# Exploring the Effect of the COVID-19 Zoo Closure Period on Flamingo Behaviour and Enclosure Use at Two Institutions

Peter Kidd [1] , Steph Ford [1] and Paul E. Rose [1,2,*]

1 Centre for Research in Animal Behaviour, College of Life & Environmental Sciences, University of Exeter, Exeter EX4 4QG, UK; peterjosephkidd@gmail.com (P.K.); stephfordmusic@hotmail.co.uk (S.F.)
2 WWT, Slimbridge Wetland Centre, Slimbridge GL2 7BT, UK
* Correspondence: p.rose@exeter.ac.uk

**Simple Summary:** The COVID-19 pandemic led to widespread closures of zoos, which afforded an opportunity to compare the behaviour of two species of flamingo at two zoos between a period of visitor absence to a period of normal visitor presence. We found that the behaviours performed and the enclosure areas used by the flamingos were consistent over time, indicating that the return of visitors did not unduly alarm the birds. One of the flamingo flocks showed minor changes in some behaviours but these effects were explained by analysis of weather influences on flamingo behaviour and visitor numbers. The constant interaction with zookeepers and the gradual opening to visitors are likely to have positively influenced the birds' behaviour and responses to the zoos' re-openings.

**Abstract:** Visitors can influence the behaviour of zoo animals through their auditory and visual presence, with mixed findings of negative, neutral, and positive effects on welfare. This study opportunistically utilised the UK-wide COVID-19 period of zoo closure to investigate the activity and enclosure usage of Greater (*Phoenicopterus roseus*) and Chilean (*P. chilensis*) Flamingos housed at two zoos. Flamingo behaviour at both sites was observed during the last week of a three-month closure period and the immediate reopening of the zoos. Photographic data were collected at three timepoints during each observation day. Negative binomial GLMMs compared the behaviour observed during zoo closure to the behaviour observed during zoo reopening, whilst accounting for climatic variables and time of day. Spearman's correlation identified relationships between behaviour with the number of visitors and weather. Greater Flamingos were not influenced by the reintroduction of visitors to the zoo setting. Chilean Flamingos showed an increase in inactivity and decrease in movement and feeding when the zoo reopened. These possible behavioural responses are better explained by the influence of temperature on the behaviour of Chilean Flamingos and by the correlation between temperature and visitor number, rather than a direct consequence of visitor presence. This research details the multifactorial nature of any potential anthropogenic effects on zoo animal behaviour and highlights the importance of considering environmental variables alongside the measurement of visitor presence or absence.

**Keywords:** COVID-19; visitors; behaviour; enclosure usage; welfare; flamingos; activity-budgets

## 1. Introduction

The behaviour of zoo animals can be indicative of responses to captive conditions, and subsequently welfare [1]. As an individual's behaviour and environment are inextricably linked, it is likely that a zoo's visitors influence the behaviour and welfare of the animal population due to their auditory and visual impact [2,3]. Coronavirus Disease 2019 (COVID-19) led to the global disruption and dramatic closure of global zoological institutions, with visitors unable to access animal exhibits for several months [4]. As such, COVID-19 provided a unique opportunity to study the influence of visitors on welfare

by measuring and comparing behaviour of captive animals during visitor absence and visitor presence.

Zoo visitors are a constant presence in the lives of zoo animals, and the effect of visitors on behaviour has been explored across a wide range of taxa [1]. Visitors can influence the behaviour (and potentially the welfare) of captive animals through disruptions to the sensory environment [2,3]. These influences have been categorised into negative, positive, or neutral effects on welfare [5,6]. Alongside changes in behavioural repertoires, investigating how an animal uses its enclosure is a commonly used determination of welfare and enclosure suitability [7]. Both a reduction of space use and an increase in retreat responses are good indicators of increased discomfort, physically or psychologically [8,9].

Responses to visitors during normal zoo opening hours are complex, both between and within avian species [1]. For instance, visitors disrupted the behaviour of captive African Penguins (*Spheniscus demersus*) resulting in decreased pool use [10], whereas Gentoo Penguins (*Pygoscelis papua*) showed increased pool usage and behavioural diversity when visitor number increased [11]. Rose et al. [7] also found that changes in visitor number and subsequent noise led to increased enclosure use, but that there was a complex relationship with climatic effects that varied between captive Greater (*Phoenicopterus roseus*) and Chilean (*P. chilensis*) Flamingos. Retreat responses, whereby animals move to areas of the enclosure further from visitors, have been documented in 24 bird species in a walk-through aviary [12] while no effect of visitor presence was found in a pair of Black-casqued Hornbills (*Ceratogymna atrata*) [13] or in flocks of Greater, Caribbean (*Phoenicopterus ruber*), Chilean, Andean (*Phoenicoparrus andinus*), and Lesser (*Phoeniconaias minor*) Flamingos [14].

Studies have also investigated responses to visitors by observing and comparing behaviour between periods where visitors are not allowed at an enclosure or zoo to periods of normal visitation. Where birds have been studied, findings have again been mixed. A study of Greater Rhea (*Rhea americana*) found no significant differences in behaviour upon visitor return, suggesting that habituation to human visitors was maintained during absence and prevented behavioural change at the point of visitor return [15]. Captive Little Penguins (*Eudyptula minor*) were shown to display increased aggression, huddling, and avoidance behaviours when visitors were present [16], whereas no changes in the behaviour of African Penguins were found after the reintroduction of visitors [4].

In other taxa, visitor deprivation increased vigilance, reduced interaction with their environment and increased retreat responses in meerkats (*Suricata suricatta*), but also increased positive visitor interactions [4]. Grevy's zebra (*Equus grevyi*) showed increased comfort behaviours and increased use of zones closest to public viewing areas when visitors were not present [17]. Chinese goral (*Naemorhedus griseus*) also showed increased interaction with their physical environment when visitors were not present [17].

The effect of visitors on the behaviour and zone usage of zoo animals is therefore complex, with effects varying between related species, and findings that are difficult to disentangle from the influence of enclosure design and climate. A research bias towards mammals and megafauna [18] also hinders our understanding of visitor effects in avian species and has led to a gap in husbandry and welfare literature for birds, despite their ubiquitous presence in zoos [18]. A review by Sherwen and Hemsworth [6] found that only 8% of visitor effect studies focussed on birds. To advance avian research in zoological collections, flamingos are excellent species for behavioural studies due to their widespread presence in zoos, large sample sizes, gregariousness, diverse behavioural repertoires, and visitor popularity [19]. Greater and Chilean Flamingos are both omnivorous, shallow-keeled (a bill containing wider lamellae to filter for larger aquatic food items) flamingos that utilise saline wetlands and salt flats for feeding and nesting [20,21]. Greater Flamingos are the largest and most widespread flamingo species [22], occurring in a range of wetland environments. Chilean Flamingos are smaller and more brightly coloured, with distinctive pink joints on grey legs [23]. Chilean Flamingos occur predominantly in wetlands along the western side of South America and are declining in number in the wild due to a range of anthropogenic threats [24]. We suggest that Greater Flamingos may be more adaptable to a

changing presence of zoo visitors due to their more generalised ecology when compared to Chilean Flamingos. As such, this study aims to continue to add to the non-mammalian literature on visitor effects by examining the behaviour and enclosure usage of these two commonly housed flamingo species and their response to the resumption of zoo visitors after COVID-19 enforced closure at two UK zoological collections. Given that research has reported variation by species, population and location of how wild bird behaviour was impacted by COVID-19 lockdowns [25,26], we wished to understand whether similar influences on behaviour may occur in captive populations.

## 2. Materials and Methods

### 2.1. Study Species and Study Sites

The behaviour and enclosure usage of Chilean Flamingos (N = 37) housed at Zoological Society of East Anglia (ZSEA) Banham Zoo, Norfolk, UK, and Greater Flamingos (N = 36) held Africa Alive, Suffolk, UK (ZSEA), were observed during COVID-19 zoo closure and subsequent zoo-reopening. Specific sampling period closure periods for each zoo are detailed in Table 1. Enclosure zones are illustrated in Figure 1 with further details provided in the Supplementary Materials (Tables S1 and S2). Demographic information on each flamingo was provided by each zoo. Husbandry routines (including cleaning of indoor housing and management of pool water quality) at each enclosure were similar and both species were provisioned during opening and closure. Flamingos were fed early in the morning on a bespoke flamingo pellet and keepers would also check on the birds throughout the course of the day, re-filling feeding bowls if needed. Such interruptions were factored into data collection schedules. Both flamingo enclosures were mixed species, with flamingos being housed with various wildfowl (e.g., geese *Answer and Branta* sp., whistling ducks, *Drendocygna* sp.) and/or Sacred Ibis (*Threskiornis aethiopicus*), White Stork (*Ciconia ciconia*), Demoiselle Crane (*Grus virgo*) and Helmeted Guineafowl (*Numida meleagris*). No adverse interactions between these birds and either species of flamingo were noted during data collection.

**Table 1.** Details of the data collection schedule for each flamingo species and total number of visitors on site (Gate Admissions).

| Study Enclosure Site (Size) | Study Species (N) | Period of Data Collection | | |
| | | Zoo Closed | Zoo Open | |
| | | Dates | Dates | Gate Admissions (Zoo Open Dates Only) |
|---|---|---|---|---|
| Africa Alive, Suffolk, UK (2868 m²) | Greater Flamingo (N = 36) | 25/06/2020 | 29/06/2020 | 200 |
| | | 26/06/2020 | 01/07/2020 | 563 |
| | | 27/06/2020 | 02/07/2020 | 384 |
| | | 28/06/2020 | 03/07/2020 | 544 |
| | | 30/06/2020 | 04/07/2020 | 1032 |
| | | | 06/07/2020 | 693 |
| | | | 07/07/2020 | 746 |
| | | | 08/07/2020 | 558 |
| Banham Zoo, Norfolk, UK (2118 m²) | Chilean Flamingo (N = 37) | 24/06/2020 | 28/06/2020 | 283 |
| | | 25/06/2020 | 01/07/2020 | 904 |
| | | 26/06/2020 | 02/07/2020 | 593 |
| | | 27/06/2020 | 03/07/2020 | 1032 |
| | | 29/06/2020 | 04/07/2020 | 1754 |
| | | 30/06/2020 | 06/07/2020 | 1296 |
| | | | 07/07/2020 | 1097 |
| | | | 08/07/2020 | 868 |

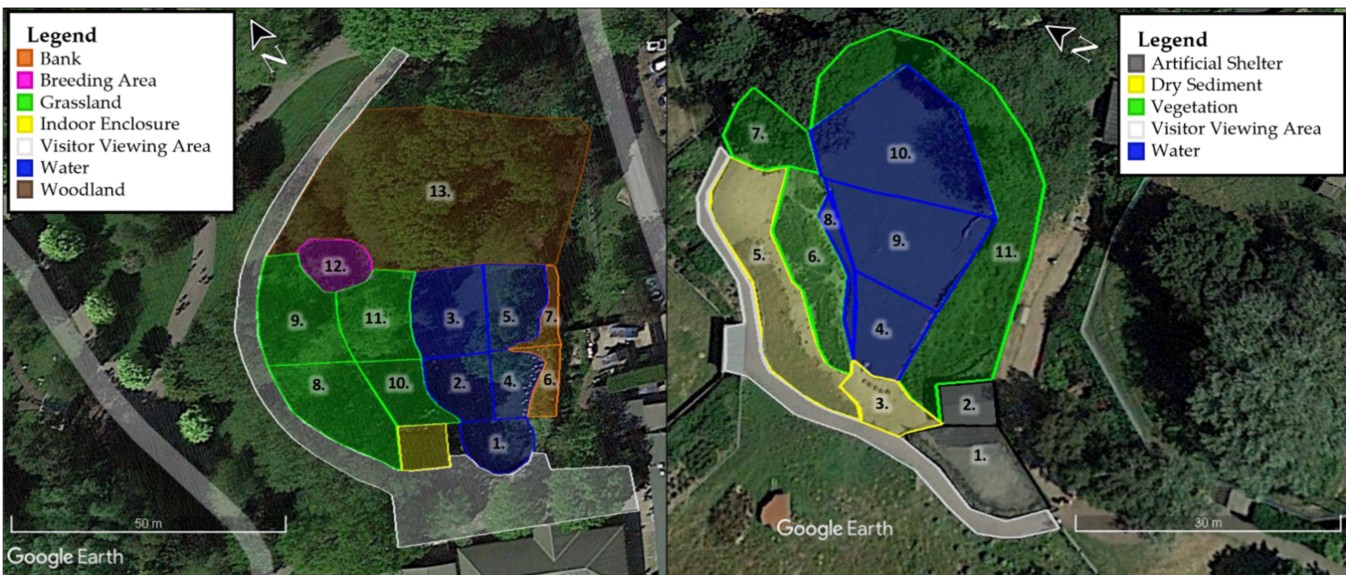

**Figure 1.** Map data ©2021 Google Earth Pro. Left: Greater Flamingo enclosure at Africa Alive. 1 = Overlooked; 2 = Front Left Water (FLW); 3 = Back Left Water (BLW); 4 = Front Right Water (FRW); 5 = Back Right Water (BRW); 6 = Front High Bank (FHB); 7 = Back Low Bank (BLB); 8 = Back Grassland Path (BGP); 9 = Front Grassland Path (FGP); 10 = Front Grassland Water (FGW); 11 = Back Grassland Water (BGW); 12 = Nesting Zone (NZ); 13 = Woodland. Right: Chilean Flamingo enclosure at Banham Zoo. 1 = Yard; 2 = Shelter; 3 = Beach; 4 = Front Shallow Water (FSW); 5 = Hill; 6 = Slope; 7 = Canopy Cover (CC); 8 = Left Shallow Water (LSW); 9 = Middle Deep Water (MDW); 10 = Far Deep Water (FDW); 11 = Overgrown Bank (OB).

*2.2. Data Collection*

Methods for behavioural observations were identical for each flamingo flock. Prior to data collection, both zoos had been fully closed to visitors from 21 March 2020 to 1 July 2020. Prior to data collection commencing, Africa Alive had been fully closed to visitors from 21 March 2020 to 28 June 2020 with a soft opening on 29 June 2020. Banham Zoo had been fully closed from 21 March 2020 to 27 June 2020 with a soft opening on 28 June 2020. Both zoos allowed visitation by zoo members from 1 to 3 July 2020 and fully reopened to the public from 4 July 2020. Data were collected on the final week of zoo closure and the first week zoo-reopening. Researchers were randomly assigned to a zoo on each day of data collection. The time of observation was dictated by the ability of the researchers to access each zoo during lockdown and the time needed to develop methods and obtain ethical approval during a difficult working period. Researchers limited any influence on flock behaviour by arriving at the data collection position at least 15 min before scan sampling started. The auditory and visual presence of the researchers were kept to a minimum during data collection. All data were collected by SF and PK. To ensure the reliability of recorded behaviours, PK trained SF in identification of flamingo behaviours due to previous experience with this data collection method. Photos of bird behaviour were reviewed by each researcher throughout the project to ensure there was no drift from the ethogram.

On each day of data collection, both species were observed during the morning (10:00–11:30), early afternoon (12:00–13:30) and late afternoon (14:45–16:15) sampling periods. Time slots were chosen based on when researchers could access the birds at each zoo to standardise observation times at each flamingo flock. During each observation period, an instantaneous scan sampling method [27] was employed with behaviour recorded from photographs of the flamingo as per the method described in Rose et al. [14]. Flock-wide photographs were taken at 60 s intervals using digital cameras (Canon EOS 750D Digital SLR; Panasonic LumixG Digital). Observations were paused when a zookeeper entered the enclosure to carry out routine husbandry, beyond the public viewing areas. Observations resumed 15 min after the zookeeper had left the enclosure.

During the zoo-reopening period, the number of visitors visible to the researcher in the enclosure was recorded every 60 s. Records of daily gate admissions were also provided by each zoo (Table 1). Local weather measurements of temperature (°C), wind speed (km/h), precipitation (mm), sunshine (%), and humidity (%) were retrospectively recorded from worldweatheronline.com [28].

### 2.3. Image Processing

Using a predetermined ethogram (Table 2) adapted from Rose [29] the photographic data were manually assessed for behaviour. The number of key behaviours performed by flamingos within the flock was counted and summed for each 60 s sample taken. Images were also assessed for enclosure usage. Each enclosure was separated into zones based upon biologically relevant features and proximity to visitor viewing areas (Figure 1). The number of flamingos within each zone was counted and summed for each 60 s sample taken. Counts of behaviour and enclosure usage were entered into a Microsoft Excel database. Images from both zoos were processed in the same manner.

**Table 2.** Ethogram of flamingo state behaviours.

| Behaviour | Definition |
|---|---|
| Preening | Cleaning feathers with bill or water by scooping water over the body with wings and or bill, either sitting or standing. Flamingo uses feet or head to move plumage behind its head. |
| Foraging | Trough feeding (Consumption of food from feeders provided by zoo: head is either inside trough or directly above trough); Natural feeding (Moving the bill through the water from side to side or picking food off the ground with bill. Can occur either whilst still or moving. Flamingo is extending its neck down to the floor, either whilst standing, walking, or sitting) |
| Inactive | Rest (Motionless with head tucked under wing, standing, or sitting, with eye(s) open or closed); Standing (motionless, not vigilant, head is held in front of body. General inactivity, with neck in a relaxed 'S' shape. Bill is pointed at the ground); Sitting (Flamingo is sitting motionless, not vigilant. Eyes are open, legs are tucked under body and neck is not erect). |
| Aggression | Spreading the scapular feathers to look more threatening, either whilst sitting or standing; Hooking and/or jousting (extension of the neck either fully straight or in an extended S-bend, and pointing of the bill at a nearby bird with the head swayed side to side and engaging in direct contact, spreading of scapular feathers; Fighting (birds push and shove one-another, using wings and beaks, either sitting or standing). |
| Courtship | Head flagging (neck straight and erect, head is held above a 90 degree ankle, jerked from side to side quickly); Marching displays (birds pack closely together and move quickly in an exaggerated fashion with straight heads and necks); Wing salute (wings are opened quickly and then snapped shut, whilst standing.); Wing-leg stretch (one wing is outstretched along the leg, which is also being stretched on that side of the body); Twist preen (wing is opened up and outwards and to the side, but not fully extended and the head and bill are placed behind the opened wing as if preening its black primary feathers.); Mating attempt (copulation between male and female birds). |
| Movement | Running/walking (bipedal movement along the ground at a hurried /gentle pace. Identifiable as the 'peeling' of one foot off the ground, or the foot in mid-air extended forward. Includes wading through water. If running, wings may also be open); Swimming (movement across the water similar to a duck); Social following (positive social interaction where one flamingo follows another directly behind across the enclosure). |
| Vigilance | Neck held in an erect 'S' shape or fully extended, whilst visually scanning surroundings, either sitting or standing. Bill is held up slightly compared to relaxed standing posture. |
| Unknown | Bird is out of sight or performing a behaviour not described in this ethogram. |

The modified Spread of Participation Index (SPI) [30] was used to determine use of available space, given as: SPI $= \Sigma \, |fo - fe| / 2(N - f\text{min})$; where $fo$ is the observed frequency of observations in a zone, $fe$ the expected frequency of observations in a zone based on zone size and even use of the whole enclosure, N the total number of observations in all zones and $f\text{min}$ the expected frequency of observations in the smallest zone. 0 suggests equal use of all zones, and 1 suggests exclusive use of only one zone.

*2.4. Statistical Analysis*

A total of 52 h of data were collected on the Greater Flamingo flock and 53 h of data on the Chilean Flamingo flock. Behavioural and zone usage data were analysed using RStudio v.1.3.1073 [31]. Data exploration indicated that dependent variables were overdispersed, violating the assumption of a Poisson distribution. Using the lme4::glmer package (v1.1-26) [32] we fitted full conditional Generalized Linear Mixed Models (GLMMs) with a negative binomial error distribution. Each model was fitted with zoo status (with two levels: closed, open) and observation period (with three levels: morning, early afternoon, late afternoon) as categorical predictors. To control for the effect of climate, temperature, wind speed, precipitation, humidity, and sunshine were entered as continuous predictors. The date was added as a random intercept to allow for behavioural responses to vary over time.

Several models explaining the activity and zone usage in both flamingo populations did not converge. Variance Inflation Factors (VIFs) also indicated that several models contained over-dispersed variables (i.e., VIFs $\geq$ 5). All continuous variables were centred to reduce multicollinearity. Where this did not reduce VIFs, highly correlated independent variables were removed using a stepwise procedure until all VIFs were below the cut-off point (VIF < 5). Models were then checked for singularity using the performance::check_singularity package [33]. All final models were non-singular. Supplementary Materials explain the outputs of the model fit and term selection process (Tables S3–S10).

The coefficient of determination ($r^2$) adjusted for mixed models was calculated using the MuMIn::r.squaredGLMM package [34] to determine the proportion of variation in the behavioural responses explained by each model. We then used a stepwise procedure using lme4::drop1 [32] to determine the singular influence of each parameter within each model. The relationship between each predictor and outcome were inferred through odds ratios, taken as the exponent of beta coefficient estimates. Odds ratios represent the probability that an outcome of interest occurs given a particular exposure, compared to the odds of the outcome occurring in the absence of the given exposure [35]. This allows for an easy interpretation to compare the likelihood of behavioural outcomes occurring given the exposure of visitors (zoo open), compared to when behaviour occurs in the absence of visitors.

Data were then split by zoo status for analysis on zoo open data only. Spearman correlations were run to test for associations between visitor number (at the enclosure) and activity or zone occupancy. Spearman correlations were also run to test for associations between visitor number and climatic variables. Where behavioural responses correlated with visitor number, negative binomial GLMMs were run. Interactive terms between climatic variables and visitor numbers were entered into the models to account for the relationships where climatic factors previously correlated with visitor numbers. The main effect of visitor number could then be assessed independently of the effect that weather had on the number of visitors entering each zoo.

## 3. Results

*3.1. Flamingo Behaviour*

Time spent on each behaviour did not significantly differ between zoo closure and zoo open conditions for Greater Flamingos (model output in Supplementary Materials; Table S11). Chilean Flamingos fed less and were more inactive when the zoo was open compared to when the zoo was closed (feeding: mean$_{closed}$ = 15.6%, SE = $\pm$1.012, mean$_{open}$ = 5.7%, SE = $\pm$0.538; inactive: mean$_{closed}$ = 38.2%, SE = $\pm$1.356, mean$_{open}$ = 47.2%, SE = $\pm$1.164). Differences in Chilean Flamingo behaviour with zoo status are shown in the model output in the Supplementary Materials (Table S12). Flamingo behaviour between periods of zoo closure and opening for both species are shown in Figure 2. The average % time spent on all behaviours by time of day and by zoo status is contained in the Supplementary Materials (Tables S13 and S14).

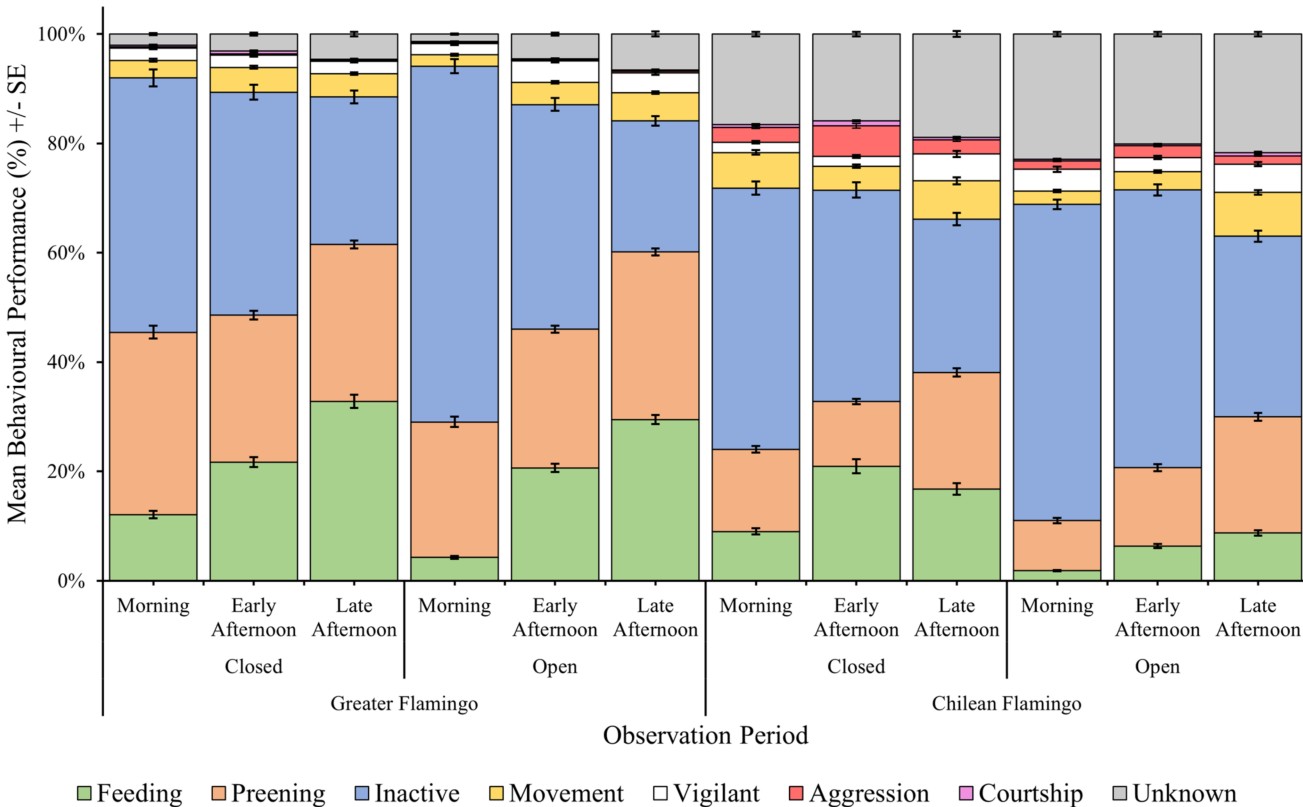

**Figure 2.** Average time-activity budgets of Greater and Chilean Flamingo flocks across observation periods of zoo status. Error bars represent the variation (SE±) in mean performance of behaviours. Greater Flamingos spent most of their time inactive, preening, and feeding. Chilean Flamingos spent most of their time inactive, preening, feeding, or their behaviour was unknown. Feeding decreased and inactivity increased in the Chilean Flamingo flock when the zoo reopened.

The effect of zoo status, observation period, temperature, wind, rainfall, humidity, and sunshine with the random intercept explained 0.02–44.38% and 1.90–44.19% of the variance for each behaviour of the Greater and Chilean Flamingo flocks, respectively (see model fit values in the Supplementary Materials; Tables S3 and S4).

### 3.1.1. Greater Flamingo Behaviour

Zoo status did not significantly improve model fit when explaining feeding, preening, inactivity, movement, vigilance, aggression, courtship, or unknown behaviour, $X^2$'s (1, N = 2999) $\leq$ 2.158, $p$'s $\geq$ 0.151. Observation period significantly improved model fit for models explaining feeding, preening, inactivity, movement, vigilance, and unknown behaviours $X^2$'s (2, N = 2999) $\geq$ 11.069, $p$'s $<$ 0.001. Changes in model fit values through stepwise deletion of singular parameters, as well as fixed factor outputs for behaviour are displayed in the Supplementary Materials (Tables S7 and S11) and visually represented in Figure 3.

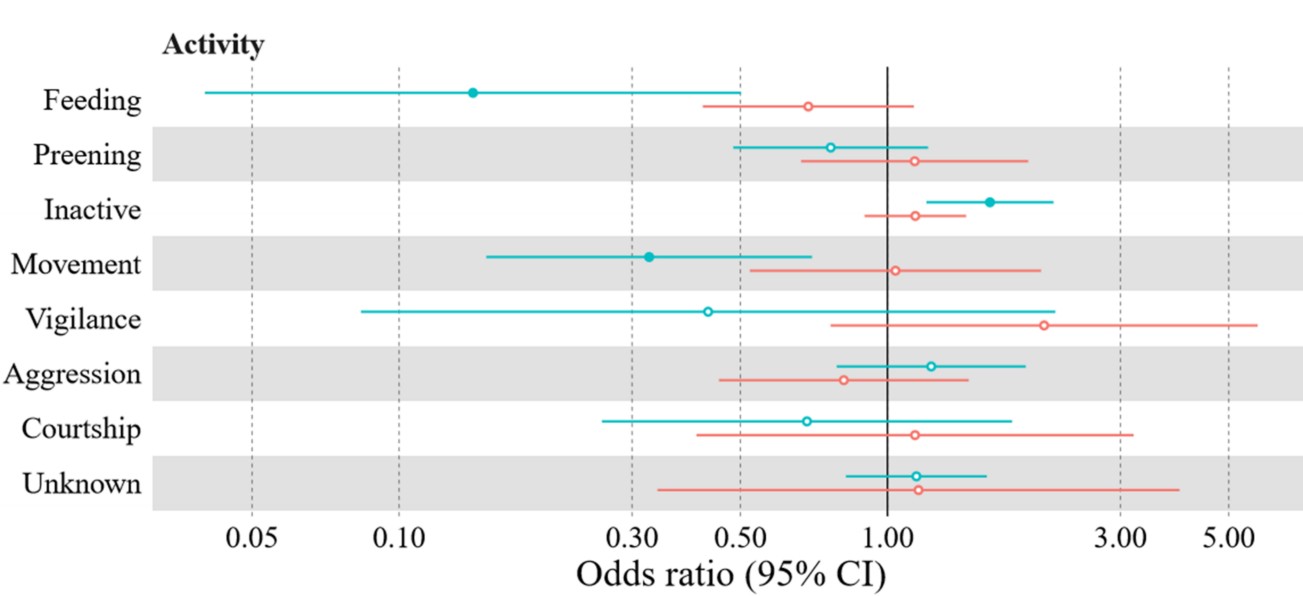

**Figure 3.** Odds ratios and 95% confidence intervals of the Greater and Chilean Flamingo flock to perform key behaviours when the zoos were open, compared to when the zoos were closed. Intercept = zoo closed. Hollow points = $p \geq 0.05$; filled points = $p < 0.05$. The Greater Flamingos were just as likely to perform key behaviours when the zoo was open than when the zoo was closed. Compared to when the zoo was closed, Chilean Flamingos were 1.622 times more likely to be inactive, 0.142 times less likely to be feeding, and 0.325 times less likely to be moving when the zoo was open.

Results of Spearman's correlations between visitor number and key behaviours performed during zoo open condition indicated no significant associations, $r_s$'s (1817) = $-0.099$–$0.083$, $p$'s < 0.341. Where a test statistic is followed with an "'s", this indicates that multiple outputs are being reported at once and the reader should refer to the referenced tables and Supplementary Materials to see individual test outputs for specific dependent variables. Correlations between behaviour and environmental factors are displayed in the Supplementary Materials (Table S15) and visually presented in Figure 4. Correlations between a visitor and climatic variables are displayed in Table 3.

3.1.2. Chilean Flamingo Behaviour

Zoo status significantly improved model fit when explaining feeding, inactivity, and movement behaviours in the flock of Chilean Flamingos, $X^2$ (1, N = 3198) $\geq 6.042$, $p < 0.05$. When Banham Zoo reopened, the flock of Chilean Flamingos were more likely to be inactive, $\beta = 0.483$, SE = 0.153, $p < 0.01$, OR = 1.622 (95% CI: 1.202, 2.188), less likely to be moving, $\beta = -1.124$, SE = 0.392, $p < 0.01$, OR = 0.325 (95% CI: 0.151, 0.701) and less likely to be feeding, $\beta = -1.954$, SE = 0.645, $p < 0.01$, OR = 0.142 (95% CI: 0.030, 0.502). Observation period also significantly improved model fit for models explaining feeding, preening, inactivity, movement, vigilance, aggression, and unknown behaviours, $X^2$ (2, N = 3198) $\geq 13.178$, $p < 0.01$. Change in model fit values through stepwise deletion of singular parameters, as well as fixed factor outputs for activity are displayed in the Supplementary Materials (Tables S8 and S12) and visually represented in Figure 3.

Results from Spearman correlations between visitor number and key behaviours performed during zoo open condition indicated a weak significant negative association with vigilance, $r_s$(1915) = $-0.183$, $p < 0.001$. When controlling for the relationship between climate and visitors in GLMMs, visitor number maintained a significant negative relationship with vigilance, $\beta = -0.16$, SE = 0.07, $p < 0.05$. Correlations between behaviour and environmental factors are displayed in the Supplementary Materials (Table S16) and

visually presented in Figure 5. Correlations between visitor and climatic variables are displayed in Table 3.

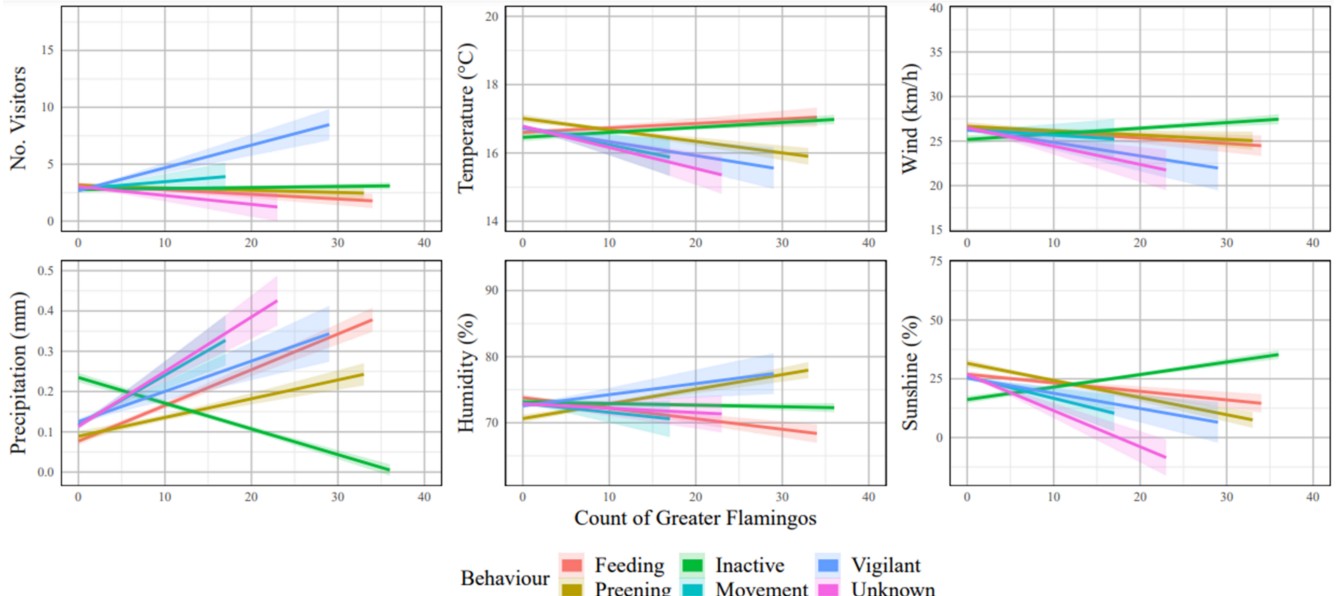

**Figure 4.** Correlation between the counts of key behaviours performed by the greater flamingo flock during zoo-reopen condiiton and enviornmental factors. Number of visitors did not correlate with behaviour. Shaded areas around lines represent ±95% CI. Weak signifcant correlations (0.1 ≤ $r_s$ ≤ 0.29) were held between: temperature-inactivity, unknown; wind-inactivty, vigialnce, and unknown; precipitation-feeding, movement, vigilance, and unknown; humidity-preening and inactivity; sunshine-preening, inactivity, movement, vigilance, and unknown. A significant moderate correlation (0.3 ≤ $r_s$ ≤ 0.49) was shown between precipitation and inactivity.

**Table 3.** Spearman's rank correlation outputs testing association between climatic variables and the number of visitors within the Greater and Chilean Flamingo enclosure during zoo re-open condition.

| Correlates | Greater Flamingo | | Chilean Flamingo | |
|---|---|---|---|---|
| | No. Visitors | | No. Visitors | |
| | **rho** | ***p*** | **rho** | ***p*** |
| Temp | 0.123 | <0.001 | 0.129 | <0.001 |
| Wind | 0.032 | 0.167 | 0.084 | <0.001 |
| Precipitation | 0.100 | <0.001 | −0.167 | <0.001 |
| Humidity | 0.129 | <0.001 | 0.230 | <0.001 |
| Sunshine | −0.084 | <0.001 | −0.206 | <0.001 |

No correlation: rho = ±0–0.09; weak correlation: rho = ±0.10–0.29; moderate correlation: rho = ±0.3–0.49; strong correlation: rho = ±0.5–1. *p* values indicate whether the association occurred by chance, they do not infer strength of association.

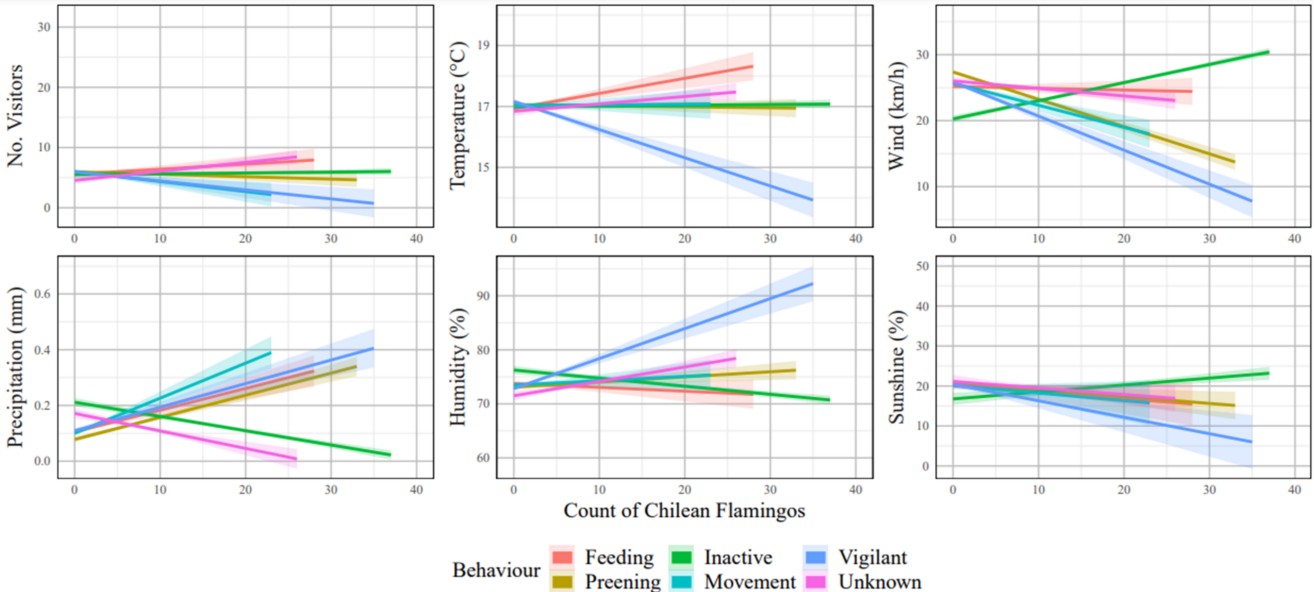

**Figure 5.** Correlation between counts of key behaviours performed by the Chilean Flamingo flock during the zoo-reopen condition and environmental factors. Shaded areas around lines represent ±95% CI. Number of visitors showed a weak negative correlation with vigilance. Weak significant correlations ($0.1 \leq r_s \leq 0.29$) also held between: temperature-feeding, vigilance; wind-movement; precipitation-feeding, movement, vigilance, unknown; humidity-inactivity, vigilance. Significant moderate correlation ($0.3 \leq r_s \leq 0.49$) was shown between: wind-preening, inactivity, vigilance; precipitation-preening, inactivity.

### 3.2. Flamingo Enclosure Usage

Occupancy of enclosure zones did not differ between zoo closure and zoo open conditions for Greater Flamingos (see Supplementary Materials; Table S17). Chilean Flamingos showed increased usage of zone 5 (Hill) when the zoo reopened (mean$_{closed}$ = 47.3%, SE = ±1.394, mean$_{open}$ = 80.9%, SE = ±0.899). Differences in Chilean Flamingo enclosure usage with zoo status are shown in the model output in the Supplementary Materials (Table S18). The overall SPI values of both flocks were high (SPI$_{greater}$ = 0.844, SPI$_{Chilean}$ = 0.762), indicating low spread of enclosure usage. Differences in SPI values and patterns of enclosure usage are displayed in Figure 6 and zone occupancy (% time) is displayed in the Supplementary Materials (Tables S19 and S20).

The effect of zoo status, observation period, temperature, wind, rainfall, humidity, and sunshine with the random intercept explained <0.001–59.97% and <0.001–93.15% of the variance in occupancy of each zone for the Greater Flamingo and Chilean Flamingo flocks, respectively (see model fit values in the Supplementary Materials; Tables S5 and S6).

### 3.2.1. Greater Flamingo Enclosure Usage

Zoo status did not significantly improve model fit when explaining occupancy in zones 1–5, 10, 11 or when out of sight, $X^2$'s (1, N = 2999) ≤ 1.715, $p$'s ≥ 0.151. Observation period significantly improved model fit when explaining occupancy in zones 1–5 and out of sight, $X^2$'s (2, N = 2999) ≥ 201.236, $p$'s < 0.001. Changes the in model fit values through stepwise deletion of singular parameters, as well as fixed factor outputs for activity are displayed in the Supplementary Materials (Tables S9 and S17) and visually represented in Figure 7.

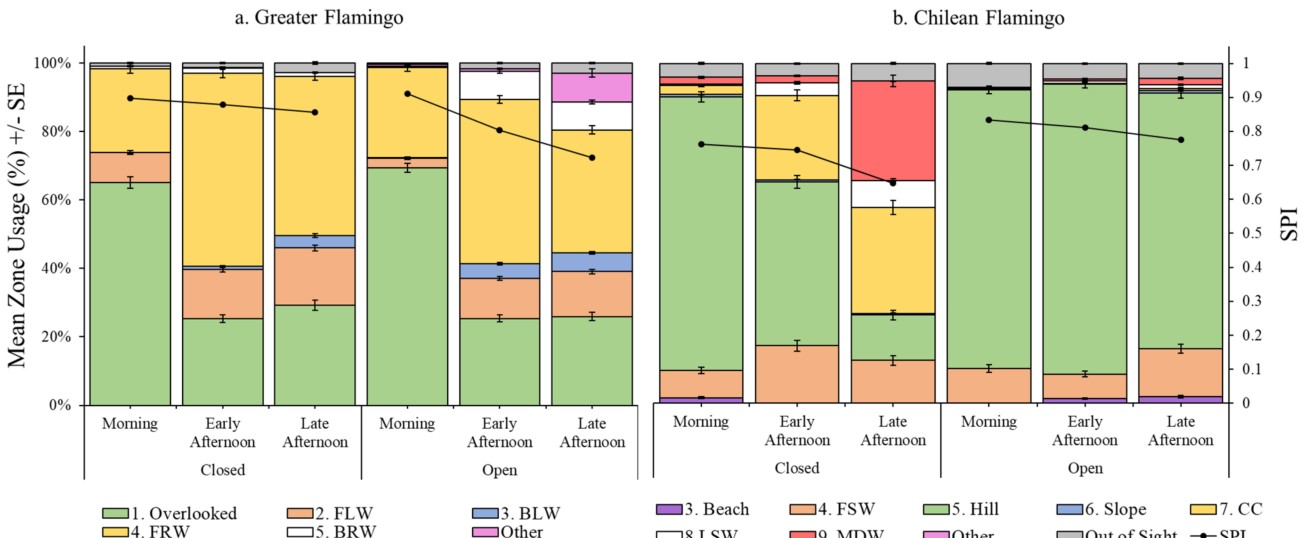

**Figure 6.** Mean zone occupancy % (stacked columns) and SPI (line plot) for the Greater and Chilean Flamingo flock across each observation period for zoo status. Error bars represent the variation (SE±) in mean occupancy. Greater Flamingos spent most of their time in zone 1 Overlooked and zone 4 FRW. Chilean Flamingos mostly occupied zone 5 Hill and increased occupancy of zone 5 Hill increased when the zoo reopened. The SPI different zoo status showed a decreasing trend for the Greater Flamingo flock and an increasing trend for the Chilean Flamingo flock. (**a**): FLW = front left water; BLW = back left water; FRW = front right water; BRW = back right water. (**b**): FSW = front shallow water; CC = canopy cover; LSW = left shallow water; MDW = middle deep water.

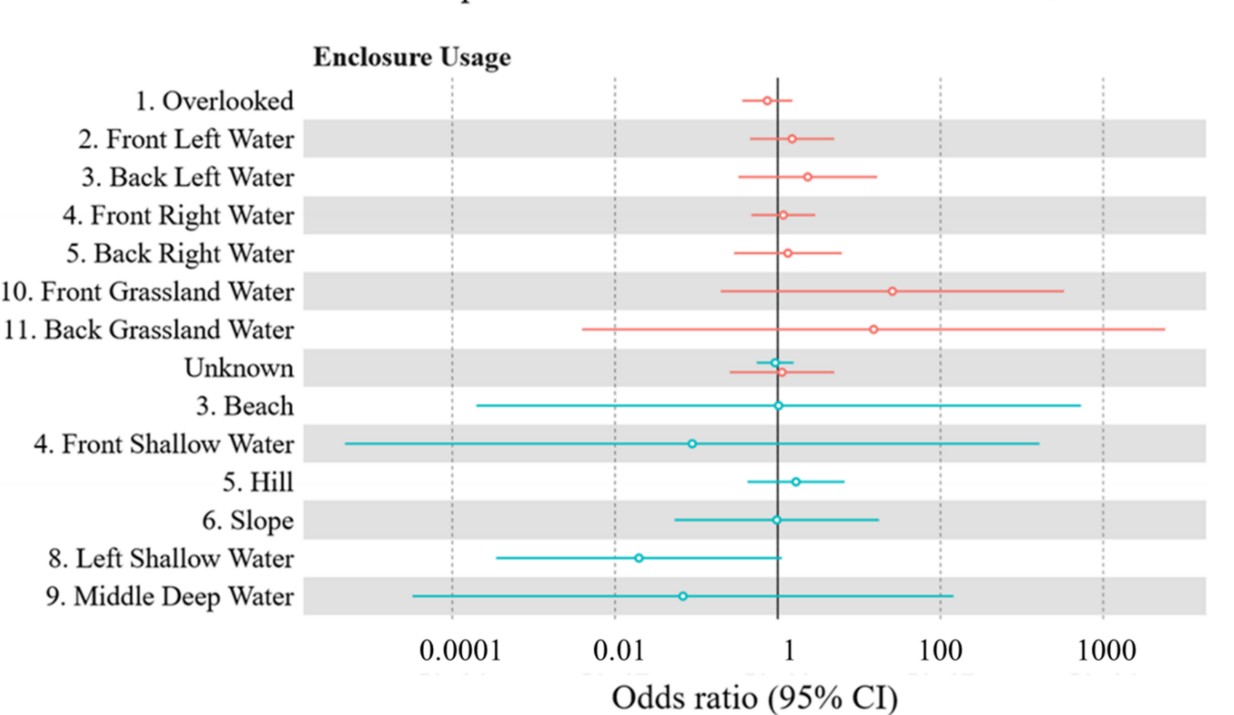

**Figure 7.** Odds ratios and 95% confidence intervals of the Greater and Chilean Flamingo flock to occupy zones of their enclosure when the zoos were open, compared to when the zoos were closed. Intercept: zoo closed. Hollow points: $p \geq 0.05$; filled points: $p < 0.05$. The odds of both flocks to occupy zones of their respective enclosures did not differ between zoo closed and zoo open conditions.

Results from Spearman correlations between visitor numbers and zone usage during zoo open condition indicated no significant associations, $r_s$'s (1817) = −0.090–0.065, $p$'s < 0.877. Correlations between SPI values and environmental factors are displayed in Figure 8. Correlations between zone usage and environmental factors are displayed in the Supplementary Materials (Table S21 and Figure S1). Correlations between visitor number and climatic variables are displayed in Table 3.

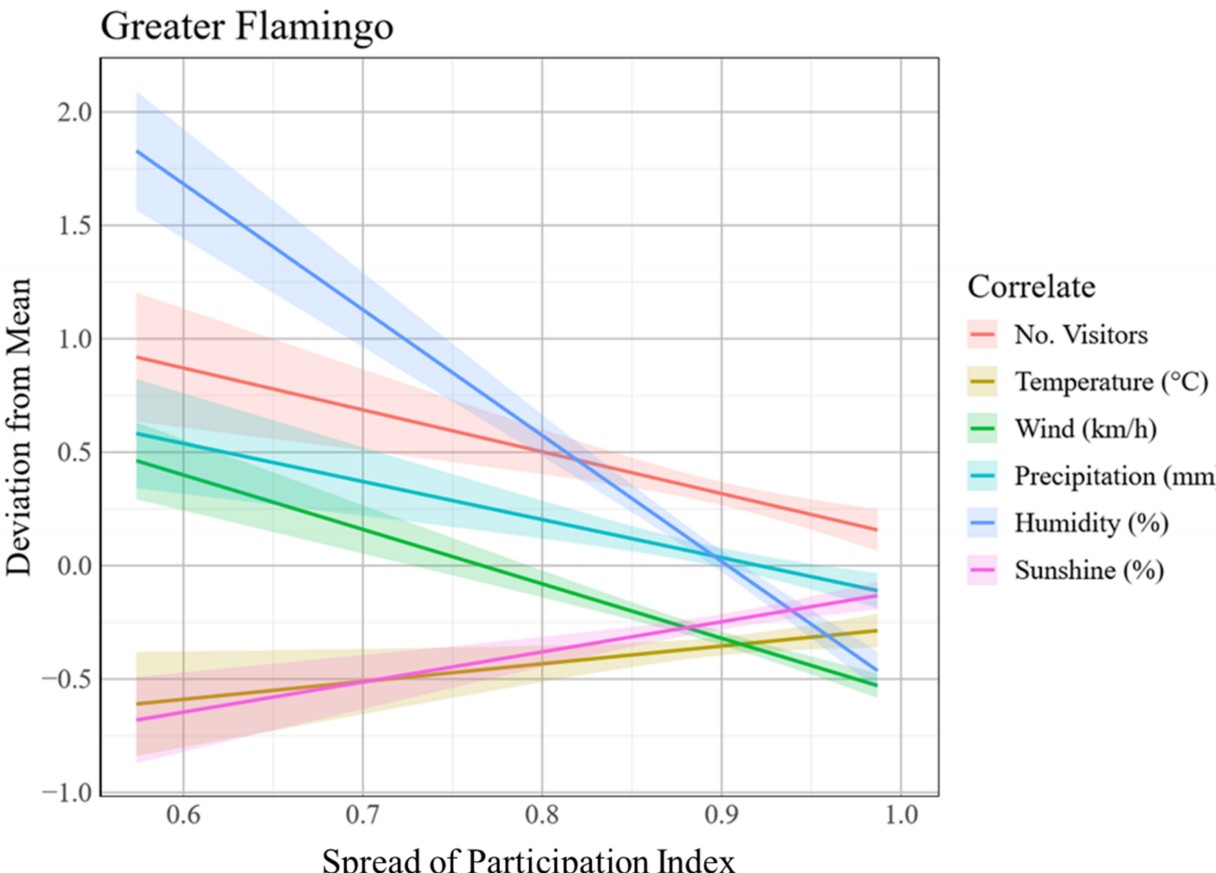

**Figure 8.** Correlation between Spread of Participation Indices (SPI) of Greater Flamingos during the zoo open condition and visitor and environmental variables. Correlates were centred around the mean. Intercept = mean. Shaded areas around lines represent ±95% CI. An increase in visitor number, wind, precipitation, and humidity was associated with a less widespread use of the enclosure. An increase in temperature and sunshine were associated with a wider use of zones.

3.2.2. Chilean Flamingo Enclosure Usage

Zoo status significantly improved model fit when explaining occupancy in zone 8, $X^2$'s (1, N = 3198) ≥ 3.627, $p$'s < 0.05. However, the fixed effect of zoo status was not significant, β = −3.929, SE = 2.063, $p$ = 0.05, OR = 0.02 (95% CI: 0.001, 1.121). Observation period significantly improved model fit when explaining occupancy in zones 3–6, 9 and when birds were out of sight, $X^2$'s (2, N = 3198) ≥ 25.914, $p$'s < 0.001. Changes in model fit values through stepwise deletion of singular parameters are displayed in the Supplementary Materials. All fixed factor outputs for enclosure usage of both flocks are displayed in the Supplementary Materials (Tables S10 and S18) and are visually represented in Figure 7.

Results from Spearman's correlations between visitor numbers and zone usage during zoo open conditions indicated a weak significant negative association with occupancy in zone 6, $r_s$(1915) = −0.147, $p$ < 0.001. When controlling for interactions between climate and visitors in GLMMs, the number of visitors did not hold a significant association with the use of zone 6, β = −0.070, SE = 0.143, $p$ = 0.055. Correlations between SPI values and environmental factors are displayed in Figure 9, and cor-

relations between visitor number and climatic variables are displayed in Table 3. Further correlations between zone usage and environmental factors are displayed in the Supplementary Materials (Table S22 and Figure S2).

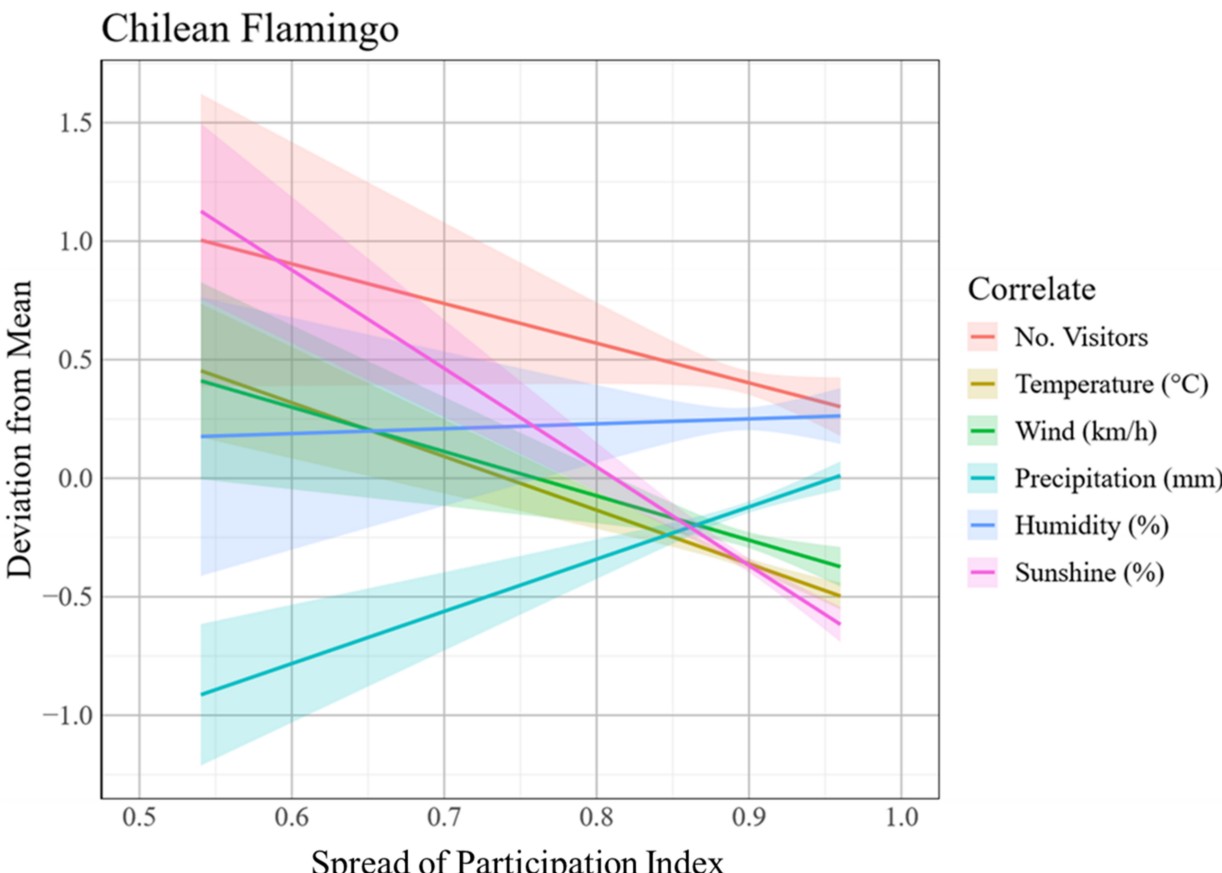

**Figure 9.** Correlation between Spread of Participation Indices (SPIs) of Chilean Flamingos during the zoo open condition and visitor and environmental variables. Correlates were centred around the mean. Intercept = mean. Shaded areas around lines represent ±95% CI. An increase in visitor numbers, temperature, wind, and sunshine were associated with less widespread use of the enclosure. An increase in precipitation and humidity were associated with a wider use of zones.

Table 4 provides an overall picture of the results generated from each flamingo flock's data.

**Table 4.** Overall summary of behavioural findings in relation to visitors and weather.

| Flock | Did Zoo Reopening Affect Behaviour? | Did Zoo Reopening Affect Enclosure Usage? | Did Increasing Visitor Numbers Show a Relationship with Behaviour? | Did Increasing Visitor Numbers Show a Relationship with Enclosure Usage? | Did Increasing Visitor Numbers Show a Relationship with Weather? | Did Flamingo Behaviour Show Any Relationship with Weather? | Did Flamingo Enclosure Usage Show Any Relationship with Weather? |
|---|---|---|---|---|---|---|---|
| Greater Flamingo | No. | No. | No. | No. | Yes, with temperature, precipitation, and humidity. | Yes, for feeding, preening, inactivity, movement, and vigilance. | Yes, for all zones analysed. |
| Chilean Flamingo | Yes, for inactivity, movement, and feeding. Inactivity increased, while movement and feeding decreased when zoo reopened. | No. | Yes, weak negative association with vigilance. Relationship maintained when controlling for weather. | No, weak negative association with occupancy in zone 6 disappeared when controlling for weather. | Yes, with temperature, precipitation, humidity, and sunshine. | Yes, for feeding, preening, inactivity, movement, and vigilance. | Yes, for all zones analysed. |

## 4. Discussion

Our results show that there were few changes in overall activity and enclosure usage of Greater and Chilean Flamingos when observing behaviour during a period of zoo closure to a subsequent reopening to visitors. The key findings from our study are summarised in Table 4. The effects of visitors on the behaviour of these two separately housed species were minimal. Both flocks of flamingos consistently performed key state behaviours and showed a biased preference toward specific zones of their enclosure during periods of both zoo closure and zoo reopening.

The inclusion of zoo status (closed vs. open) in our modelling did not improve model fit for all behaviours or enclosure usage for Greater Flamingos. Any variance in behaviour and enclosure usage was better explained by climatic variables and observation period. For Chilean Flamingos, zoo status improved model fit for feeding, movement, and inactivity, and for occupancy in zone 8. Fixed factor analysis of occupancy in zone 8 indicated that there were no effects of zoo status and the model explaining occupancy in zone 8 explained less than 0.001% of the variance (see Supplementary Materials; Table S6). This suggests that although zoo status may have explained a significant proportion of the variance, the model itself was a poor explanation of occupancy in zone 8. The effect of zoo status on behaviour was therefore limited to the Chilean Flamingo flock, and influenced only feeding, inactivity, and movement. Chilean Flamingos were more likely to be inactive, less likely to be feeding, and less likely to be moving when the zoo reopened to visitors. For all other results from the Chilean Flamingos, any variance in behaviour and enclosure usage could be better explained by weather and time of day.

Data from the zoo-reopen condition indicated that visitor number at the enclosure was not associated with changes to behaviour or enclosure usage for Greater Flamingos. For Chilean Flamingos, increased visitor number was associated with a decrease in vigilance and an increase in occupancy of zone 6. When controlling for the influence of weather on the number of visitors, the effect of zoo status on vigilance remained, whereas the main effect of visitor numbers on the occupancy of zone 6 was not significant. Taken together, our results show that alteration in visitor numbers experienced by Greater and Chilean Flamingos was associated with few behavioural changes in their activity and enclosure usage. Both flocks displayed activity budgets potentially analogous to their wild [36,37] and captive [14,38,39] counterparts, spending most of their time from 10:00 am to 16:30 pm resting, preening, or feeding [40] (although limited feeding activity in Chilean Flamingos is discussed later). Both flocks showed a preference to occupy specific zones of their enclosure that were both easily visible to visitors and close to public viewing areas; Greater Flamingos, zones 1 and 4, and Chilean Flamingos zone 5. Captive flamingos are noted for a discriminative use of their environment and strong preferences for occupying specific biologically relevant zones [14,29,41].

A lack of abnormal behaviours (e.g., pacing) or heightened alert reaction to visitors coupled with a maintained occupancy of birds' preferred zones indicated that flamingos were not negatively impacted by renewed visitor presence. Other research on COVID-19 zoo closures has reported that although species' behavioural responses were variable upon the return of visitors, there was a limited observation of negative behaviours across a wide range of species [4,17,42]. Our study extends these finding to two commonly housed zoo birds and aligns with the previous literature on captive flamingos that suggest visitor presence has a limited impact on their behaviour and welfare [7,14]. Our findings are also similar to reported neutral responses to visitors in meerkats [43], ring-tailed lemurs (*Lemur catta*) [44]; red squirrels (*Sciurus vulgaris*) [45], captive felids (Felidae) [46].

The neutral effect observed within our study suggests that zoo flamingos maintain any previous habituation to humans over a prolonged period of visitor absence. Animals habituated to humans typically present fewer negative reactions to people, whereas wild populations with less previous human experience present more dramatic behavioural changes. For instance, wild Caribbean Flamingos displayed increased vigilance and decreased feeding in response to tourists [47] whereas captive flamingos were undisturbed

by visitors [14]. Habituation allows captive animals to perform normal behaviours in the presence of unfamiliar people that correspond to positive welfare states [48]. This then allows visitors to observe and learn about biologically relevant behaviours that wild animals perform, helping zoos to achieve their educational goals [49]. It is therefore desirable for zoo animals to be habituated to the constant presence of visitors.

It is likely that the management procedures at both zoos enabled these flamingos to cope with different levels of human presence. Zookeepers develop relationships with the animals they care for and their constant presence will influence how animals cope with stressors and behave in the zoo [6]. During zoo closure, flamingos maintained their daily interactions with zookeepers who continued to care for the birds. Upon reopening, the zoos also employed a 'soft opening' with reduced visitation levels. The regular contact with zookeepers plus this gradual public opening would have minimised any risk of an adverse visitor effect.

Although overall differences in behaviour between the conditions of zoo status were minimal, some effects were apparent. The Chilean Flamingo flock displayed behavioural patterns of reduced movement and feeding, and increased inactivity when the zoo reopened to visitors. Similar patterns of behaviour in response to visitors have been documented in wild Caribbean Flamingos that displayed a reduction in feeding behaviour in response to motorised tourist boats [47], and this equated to a 13% loss in individual daily feeding opportunities. Time spent foraging in Sanderlings (*Calidris alba*), European Oystercatchers (*Haematopus ostralegus*), Piping Plovers (*Charadrius melodus*), Sandhill Cranes (*Grus canadenisis*), and five species of wildfowl reduced when human presence increased [50–54]. Increased inactivity and decreased locomotion have also been documented in captive African and Gentoo Penguins in response to increasing numbers of visitors [55]. Comparison of data collected on the behavioural responses of other species housed in the same enclosure would be a useful research extension to further evaluate such behavioural responses of the flamingos.

Reduced daytime feeding is often accompanied by other potential negative indicators of human presence, such as occupancy of foraging patches away from humans and nearer to cover, and increased vigilance [47,50,52,54]. Our study found no evidence of any retreat response or increased vigilance by these Chilean Flamingos. Visitor numbers negatively correlated with vigilance during zoo reopening and the birds maintained their enclosure usage patterns from zoo closure to zoo reopen. It is therefore unlikely that visitors were being perceived as a source of threat such that they were negatively disrupting the behaviour patterns of the flamingos. Data collection, if possible, from the start of the lockdown period would have provided information on immediate responses to the lack of visitors and how the flamingo behaviour altered accordingly. Our results provide a snapshot from the end of lockdown and into reopening. It would be interesting to see how flamingo behaviour was altered (if at all) across the three-month closure of the zoo and therefore be able to compare the degree of response to visitor return. Zoo animals are noted as performing sudden changes in enclosure usage when visitors were absent, gradually stabilising their activity pattern as they acclimated to prevailing conditions [56]. Understanding any impacts of no visitors and disruption to the running of the zoo may also be important information to evidence bird welfare states.

Reduced feeding time in the absence of any other negative visitor effects (e.g., retreat responses or increased vigilance) can be better explained by the influence of weather on these Chilean Flamingos and on the number of visitors present at the zoo. Increased sunshine is shown to reduce the diversity of enclosure zone usage in captive Chilean Flamingos [14]. Decreased foraging during daylight hours has also been suggested as a result of hotter days, where flamingos spend more time inactive and compensate by foraging in cooler nights [57]. Nocturnal activity in flamingos is an important feature of their circadian rhythms with both captive [57] and wild flamingos [58–60] performing a significant proportion of their feeding at night and spending a large amount of time inactive diurnally [61]. Flamingos employ different patterns of foraging activity in response to

prevailing environmental conditions across a 24 h period [62,63]. Should this research have continued for longer, biometric data on bird mass and plumage colour (as examples) could have been collected to determine any change in body and feather condition associated with changes to important behaviours, such as feeding. As differences in flamingo plumage colour influence the time spent foraging at the individual and group level [63], collecting data on attributes of the birds themselves would further shed light on any long term visitor effects on behaviour and welfare.

Temperature is a good predictor of visitor presence, with visitors being more numerous on warmer days [13]. Our study found that temperature positively correlated with visitor number at each enclosure and that it predicted differences in feeding behaviour for the Chilean Flamingos, i.e., increased temperatures reduced the time spent feeding. Any variation in feeding behaviour (potentially explained by visitor presence) is better explained by temperature. As this study was conducted during summer, reduced daytime feeding and increased inactivity may be a thermoregulatory mechanism employed to conserve energy during increased temperatures [58]. The apparent impact of visitors on feeding and locomotion are likely a confound of more people at the zoo during good weather as well as the daily feeding schedules of the flamingos themselves (Figure 10). The flock is likely to have compensated for this reduced daytime feeding by foraging later in the day when it is cooler, outside of this study's observation period.

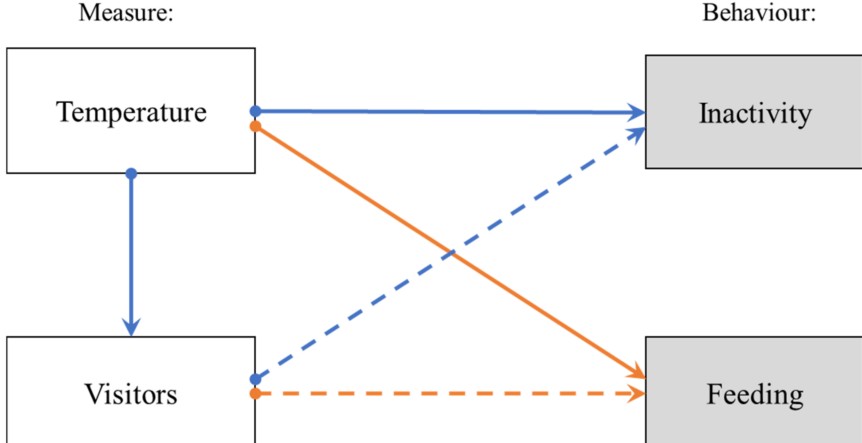

**Figure 10.** The influence of temperature on visitor numbers and flamingo behaviour. Blue: positive relationship; orange: negative relationship. Solid line: known effect; dashed line: apparent effect. Visitor presence appeared to be associated with increased inactivity and decreased feeding. However, visitor presence is itself influenced by temperature. Temperature is known to influence behaviour in the same way that visitors appeared to. The apparent effects of visitors on behaviour may therefore be due to the indirect effects of temperature. Temperature and visitor numbers are correlated and are explaining the same variation in behaviour.

It is important to note that although there was a significant reduction in feeding between zoo closure and reopening, feeding time was low during the closure period for the Chilean Flamingos (15.6% during closure). Wild Greater Flamingos spent around 51% of their daytime activity budget feeding [37] and zoo flamingos overall are noted as spending less time feeding when compared to wild birds [40]. This suggests that the birds were feeding outside of observation periods (e.g., morning or late evening) during both zoo closure and opening periods and that there was no drastic alteration to feeding behaviour when visitors returned. Such a finding is consistent weather conditions (and husbandry) being the main influences over behavioural change. Decreased time spent foraging is also influenced by food abundance with wild flamingos spending less time feeding when food biomass was greatest [62]. Food was provided abundantly by zookeepers, which helps explain the recorded foraging time in these Chilean Flamingos.

Flock-specific results for these two flamingo species are similar to published findings on other closely related species held in zoos. Behavioural responses to the return of zoo visitors differed between amphibians, suggesting species-specific and enclosure specific habituation periods to visitors are apparent [56]. African and Gentoo Penguins respond to visitors differently [10,11,55]. Although Chilean and Greater Flamingos are closely related, ecological differences exist between them. Greater Flamingos are a widespread species [22], migrating in response to seasonal changes and behavioural need, and can be observed in human-created environments such as sewage treatment works [64]. Alternatively, Chilean Flamingos inhabit coastal wetlands and remote salt lakes at elevations of ≤4500 m [24]. As such, climatic conditions may impact on Greater Flamingo behaviour less, compared to Chileans flamingos that inhabit more specialised wetlands in a more restrictive climatic range. Differences in the behavioural responses of Greater and Chilean flamingos to captive conditions are apparent in the published literature [7,14] and our findings are consistent with other research outputs. Variation in the responses to wild birds to the presence or absence of people over the COVID-19 pandemic and associated lockdowns shows how different the responses of species and populations can be to changes in the presence of humans [25,26]. For example, decreased fear responses in Tree Sparrows (*Passer montanus*) are associated with human face mask-wearing over time [65] but in other cases, prolonged face mask-wearing caused no alteration to bird behaviour or reduced fear responses across a range of species [66]. However, Mikula et al. [66] showed that habitat type was associated with differences in fear responses and therefore a similar ecological influence on response to visitors may be present in these two different species of flamingos.

We found that an increase in sunshine significantly increased the odds of the Chilean flock being inactive, whereas an increase in sunshine reduced inactivity in Greater Flamingos. Behavioural differences between these two flamingo species may have been more apparent due to the timing of data collection (during summer) but it is important to note that behavioural differences vary between populations of the same species. Time spent feeding in wild Lesser Flamingos varied between six populations whose behaviours were sampled [62]. Any behavioural differences between two separate populations are not always cause for concern but rather an example of how birds under human care are responding and adapting to the unique environments of their enclosure and at their zoo more widely.

## 5. Conclusions

Our study showed a limited to negligible visitor effect in two flocks of flamingos after a period of zero visitor presence due to COVID-19 enforced zoo closure. Our findings highlight the importance of including climatic factors when investigating potential visitor effects. We have shown that any apparent visitor effect experienced by flamingos can be better explained by environmental variables (e.g., temperature) and that it is important to consider individual species ecology when comparing any potential visitor-related impacts on welfare. Future visitor effects research should always record and evaluate local weather conditions and their effect on both animal behaviour and the number of visitors present to isolate any actual visitor-based influence on animal behaviour. Husbandry and management protocols endorsed by the two zoos involved in this study are likely to have facilitated the continued habituation of flamingos to visitors. We therefore recommend that for any future instances of zoo closure, a 'soft opening' strategy should be used during the initial period of reopening. Adopting this strategy, along with the continued activities and routines of zookeepers, has the potential to protect captive species from any negative effects of visitor presence upon reopening. Due to the global presence of captive flamingos, our research and informed practice resulting from it has the potential to protect the welfare of thousands of individual birds.

**Supplementary Materials:** The following are available online at https://www.mdpi.com/article/10.3390/birds3010009/s1. Figure S1. Correletion between counts of Greater Flamingos in enclosure zones against environmental factors. Figure S2. Correletion between counts of Chilean flamingos in enclosure zones against environmental factors. Table S1. Details of enclosure zones of the Africa Alive Greater Flamingo enclosure. Table S2. Details of enclosure zones of the Banham Zoo Chilean Flamingo enclosure. Table S3. Model fit values of negative binomial GLMMs explaining behaviour performed by Greater Flamingos at Africa Alive. Table S4. Model fit values of negative binomial GLMMs explaining behaviour performed by Chilean Flamingos at Banham Zoo. Table S5. Model fit values of negative binomial GLMMs explaining enclosure use by Greater Flamingos at Africa Alive. Table S6. Model fit values of negative binomial GLMMs explaining enclosure use by Chilean Flamingos at Banham Zoo. Table S7. Change in model fit values through stepwise deletion of singular terms within negative binomial GLMMs explaining behaviour performed by Greater Flamingos at Africa Alive. Table S8. Change in model fit values through stepwise deletion of singular terms within negative binomial GLMMs explaining behaviour performed by Chilean Flamingos at Banham Zoo. Table S9. Change in model fit values through stepwise deletion of singular terms within negative binomial GLMMs explaining zones occupied by Greater Flamingos at Africa Alive. Table S10. Change in model fit values through stepwise deletion of singular terms within negative binomial GLMMs explaining zones occupied by Chilean Flamingos at Banham Zoo. Table S11. Full conditional negative binomial GLMMs investigating the influence of predictors on the performance of key behaviours in Greater Flamingos. Table S12. Full conditional negative binomial GLMMs investigating the influence of predictors on the performance of key behaviours in Chilean Flamingos. Table S13. Average percentage of time spent performing key state behaviours by Greater Flamingos. Table S14. Average percentage of time spent performing key state behaviours by Chilean Flamingos. Table S15. Spearman's rank correlation outputs, testing for associations between climatic and visitor numbers against behaviours performed by Greater Flamingos. Table S16. Spearman's rank correlation outputs, testing for associations between climatic and visitor numbers against behaviours performed by Chilean Flamingos.Table S17. Full conditional negative binomial GLMMs investigating the influence of predictors on the occupancy of enclosure zones by Greater Flamingos. Table S18. Full conditional negative binomial GLMMs investigating the influence of predictors on the occupancy of enclosure zones by Chilean Flamingos. Table S19. Average percentage of time spent occupying enclosure zones by Greater Flamingos. Table S20. Average percentage of time spent occupying enclosure zones by Chilean Flamingos. Table S21. Spearman's rank correlation outputs, testing for associations between climatic and visitor numbers against enclosure zones occupied by Greater Flamingos. Table S22. Spearman's rank correlation outputs, testing for associations between climatic and visitor numbers against enclosure zones occupied by Chilean Flamingos.

**Author Contributions:** Conceptualization, P.K. and S.F.; Methodology, P.K., S.F. and P.E.R.; Formal analysis, P.K.; Investigation, P.K., S.F. and P.E.R.; Data curation, P.K. and S.F.; Writing—original draft preparation, P.K. and S.F.; Writing—review and editing, P.E.R.; Visualization, P.K.; Supervision, P.E.R. All authors have read and agreed to the published version of the manuscript.

**Funding:** This research received no external funding.

**Institutional Review Board Statement:** Ethical agreement was provided by the managing director of the Zoological Society for East Anglia (ZSEA) for both zoos on 19 June 2021 (Ethical framework agreement and intellectual property agreement).

**Informed Consent Statement:** Not applicable.

**Data Availability Statement:** Data can be made available upon reasonable request from the corresponding author.

**Acknowledgments:** Thank you to staff at ZSEA for their cooperation with this project, particularly at a time of such difficulty. We would like to thank Mike Woolham and Gary Batters for their support and knowledge during this study.

**Conflicts of Interest:** The authors declare no conflict of interest.

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
