# Peer review of "Exploring the Effect of the COVID-19 Zoo Closure Period on Flamingo Behaviour and Enclosure Use at Two Institutions"

_2673-6004, doi:10.3390/birds3010009_

Round 1

Reviewer 1 Report

This is a well performed and well documented study. It will add value to the existing literature on the same topic. The only minor change required in in Line 183 where the extra space break at the start of this paragraph should be removed.

Author Response

Thank you for the kind words on our manuscript. We are pleased that you feel the paper is relevant and we have actioned the edit accordingly. 

Reviewer 2 Report

This paper was a pleasure to review. The manuscript reads as if it has already been published and makes a novel, useful contribution to the scientific literature in this area. The statistical approach is spot-on, the visualizations of the data are impressive, and the conclusions are appropriate and insightful. Some suggestions for improvement are included below but are relatively minor. With some minor edits, this should be ready for publication.

  • Overall, the introduction is well written and thorough. One area of the introduction that could perhaps use some elaboration is the natural history of flamingoes. Other than describing them as gregarious and behaviorally diverse, there is not much information on the natural ecology of the two species being studied. Some of this is addressed in the discussion, but there should be some additional justification (or perhaps predictions) related to studying these particular species in the introduction (more than just that birds are understudied).
  • The formatting of Table 1 makes it hard to tell which other species are sharing enclosures with which set of flamingoes. The list just runs together. Also, the presence of other species in these enclosures is probably a confound that should be considered in data analysis (if possible) or in the discussion at least.
  • Was there a process for testing inter-observer reliability? If so, this should be included in the methods section.
  • Table 2 shows a quite extensive ethogram, and it is unclear how some of these behaviors could have been scored from photographs. Descriptions of behaviors include elements like vocalizations and swaying movements that would be impossible to discern from still photographs. The end result is that the ethogram does not read as if it is suitable for the methods, so this requires some clarification.
  • Please state how much data was collected in total.
  • How did you account for visibility in data collection? Presumably all the animals were not visible on the image for every data point. Were statistical models adjusted for animal visibility?
  • In section 3.1, activity budgets are listed in a long paragraph. This is probably too much information and perhaps leaving it in the appendix is fine. Additionally, shouldn’t there be some measure of variability (like a standard error) to go along with these means?
  • Figure 3.1 is a lot to take in. I would think about if this is the best way to display this material – is it really necessary to show time budgets for each time period, given that this was not the focus of the study? Simple bar charts comparing behavior for open vs. closed conditions separately for each species would be easier to interpret and perhaps more useful.
  • In line 228, which behavior is this referring to? Is the range given in the line above showing the range like one behavior was only explained by 0.02% while another was explained at 44.38%? If so, which behaviors performed the best and worst? This could use clarification.
  • Figures 3.2. 3.3, and 3.4 are beautiful.
  • Figure 3.5, like 3.1 is a lot to interpret and is even more challenging because the enclosure zones are not familiar to the reader. I would consider removing this and summarizing the data another way, maybe again with simple bar charts that don’t include the time of day variable.
  • The discussion is well-reasoned, and the conclusions are well supported by the data. Other than accounting for the mixed species effects, everything is covered nicely.

Minor Comments by Line

  • Line 15: should read “weather effects” I believe
  • Line 111: reads “enclosure zones and displayed” maybe a word is missing or is it “are displayed”?
  • Line 348 should refer to Figure 3.8 rather than 3.6, I believe.

Author Response

This paper was a pleasure to review. The manuscript reads as if it has already been published and makes a novel, useful contribution to the scientific literature in this area. The statistical approach is spot-on, the visualizations of the data are impressive, and the conclusions are appropriate and insightful. Some suggestions for improvement are included below but are relatively minor. With some minor edits, this should be ready for publication.

Thank you for the positive feedback and comments on the paper.

Overall, the introduction is well written and thorough. One area of the introduction that could perhaps use some elaboration is the natural history of flamingoes. Other than describing them as gregarious and behaviorally diverse, there is not much information on the natural ecology of the two species being studied. Some of this is addressed in the discussion, but there should be some additional justification (or perhaps predictions) related to studying these particular species in the introduction (more than just that birds are understudied).

Thank you for the suggestion. This has been included in the introduction to explain the ecology of the two species in more detail.

The formatting of Table 1 makes it hard to tell which other species are sharing enclosures with which set of flamingoes. The list just runs together. Also, the presence of other species in these enclosures is probably a confound that should be considered in data analysis (if possible) or in the discussion at least.

Thank you for the comment. We have edited the table and removed the list of species from the table and included this in the text. We have explained that no adverse interactions between flamingos and other birds were noted, but we have included a line in the discussion to consider future measurement of the behaviour of other species too.

Was there a process for testing inter-observer reliability? If so, this should be included in the methods section.

Thank you for comment. Details added on how the ethogram was checked for consistency.

Table 2 shows a quite extensive ethogram, and it is unclear how some of these behaviors could have been scored from photographs. Descriptions of behaviors include elements like vocalizations and swaying movements that would be impossible to discern from still photographs. The end result is that the ethogram does not read as if it is suitable for the methods, so this requires some clarification.

Thank you for the comment. The ethogram has been edited for clarity. We were not looking for vocalisation but originally kept in these explanations to give a full description of behaviour. We have edited the ethogram to ensure it is clear.

Please state how much data was collected in total.

We have included the number of hours of data collected on each flock of birds.

How did you account for visibility in data collection? Presumably all the animals were not visible on the image for every data point. Were statistical models adjusted for animal visibility?

We have accounted for this by having the unknown behavioural category and we have included unknown in the analyses.

In section 3.1, activity budgets are listed in a long paragraph. This is probably too much information and perhaps leaving it in the appendix is fine. Additionally, shouldn’t there be some measure of variability (like a standard error) to go along with these means?

Thank you for the comment. We have deleted this section as it duplicated the Figure.

Figure 3.1 is a lot to take in. I would think about if this is the best way to display this material – is it really necessary to show time budgets for each time period, given that this was not the focus of the study? Simple bar charts comparing behavior for open vs. closed conditions separately for each species would be easier to interpret and perhaps more useful.

Thank you for the comment. We have displayed the figure like this because observation period (i.e. morning, early afternoon etc) significantly improved the fit of each model, therefore we feel it is appropriate to show the behavioural change across the three observation periods as well as for zoo open and closed and for each species. We feel that this stacked bar chart provides the best way to show all data in concise manner. A normal bar chart was far too big.

In line 228, which behavior is this referring to? Is the range given in the line above showing the range like one behavior was only explained by 0.02% while another was explained at 44.38%? If so, which behaviors performed the best and worst? This could use clarification.

Edited for clarity and linked to supplementary information.

Figures 3.2. 3.3, and 3.4 are beautiful.

We are glad that you like these figures.

Figure 3.5, like 3.1 is a lot to interpret and is even more challenging because the enclosure zones are not familiar to the reader. I would consider removing this and summarizing the data another way, maybe again with simple bar charts that don’t include the time of day variable.

Thank you for the comment. We are sorry that the figure appears busy but due to the importance of time of day as an important source of variation in how flamingos use their enclosure we feel that we need to illustrate this.

The discussion is well-reasoned, and the conclusions are well supported by the data. Other than accounting for the mixed species effects, everything is covered nicely.

Thank you for the comment. We have included a sentence on mixed species.

Minor Comments by Line

Line 15: should read “weather effects” I believe

Edited

Line 111: reads “enclosure zones and displayed” maybe a word is missing or is it “are displayed”?

Edited

Line 348 should refer to Figure 3.8 rather than 3.6, I believe.

Edited

Reviewer 3 Report

I have some issues with the experimental design for this study. Birds were observed after a period of no visitors, but immediately visitation commenced. There were different acclimation periods to the no visitor/visitor conditions which make it difficult to interpret the results. The observation periods are also limited; why only three 1.5 hour periods per day? The manuscript Is not very well written or presented, making it difficult to read and interpret the results; the discussion needs to be more focusses, results needs to be removed from this section and the length needs to be reduced by 50%.

Table one: Right hand section of the table is missing. Table formatting in general is poor, making it hard to read the table. Which species are in which enclosure.

It would have been useful to observe the flamingo behaviour again after the zoo had been open for the same amount of time it had been closed. There are probably temporal effects to the behavioural responses. It doesn’t seem a “fair” comparison to observe them after they had become accustomed to no visitors, and then immediately after visitation resumed, without examining them again once they had become used to visitors. Observations immediately visitor restrictions commenced would have likewise been useful.

Figure 1 The legend is not particularly readable.

Why the 3x 1.5h observation periods per day? How were these times selected?

Paragraph line starting 194: do you mean visitor numbers into the zoo, or visitor numbers at the enclosure here?

Line 205 Performance of behaviour?

Paragraph starting line 205 This list of % is not very readable. Just summarise the major findings in the text and use the figure to show the detail.

Figure 3.1. Is the figure numbering correct?

Figure 3.1 Quality of the text is poor – it is pixilated. Suggest using a graphics package to ensure high-quality output. Can you use various forms of shading/hatching rather than colour to distinguish the various bars?

When are the animals fed relative to the observation periods?

The results section is difficult to read; work on organising the results to make the major points clear, and on writing style to aid the reader. You need to summarise your findings and make the important information clear.

Table 3.1 Why the *** for significance codes when the p values are given?

Table 4 this summary table is unnecessary and is not the way to open the discussion. Use this table to help you organise the results section – your results section should make these questions and the corresponding answers clear.

The first part to the discussion is really results.

Line 392 This sentence doesn’t make sense.

Chilean flamingos fed a lot less when there were visitors. Are there any data for body mass or body condition to examine the impact of this?

Reduction in feeding is not a negative impact?

The correlation of behaviour and temperature highlights the issue with limited sampling periods.

The length of the discussion could be reduced by 50%.

Line 480 Much of this section is really results. Beware of re-stating results in the discussion.

Formatting of references needs to be consistent (and in line with journal style). Latin names need to be in italics.

Author Response

Replies to reviewer 3

I have some issues with the experimental design for this study. Birds were observed after a period of no visitors, but immediately visitation commenced. There were different acclimation periods to the no visitor/visitor conditions which make it difficult to interpret the results. The observation periods are also limited; why only three 1.5 hour periods per day? The manuscript Is not very well written or presented, making it difficult to read and interpret the results; the discussion needs to be more focusses, results needs to be removed from this section and the length needs to be reduced by 50%.

Thank you for the helpful and developmental comments. We have addressed areas of the written style and sentence structure. We have clarified the questions about data collection. the two authors who collected data did this voluntarily. The study was planned when the lockdown started but it was a difficult time to work, zoos were stretched with resources and developing methods and getting approval needed to be right before data collection started. In an ideal world, more data would have been gathered but due to the constraints of working in a pandemic, this was not possible.

We agree that more data would be useful be we still show no effect of reopening. Had we had data at the start of lockdown, we would have results that could show flamingos responded to the onset of no visitors. As we make no inferences about this from our results, we can still present the change (or lack of) in bird behaviour from lockdown to reopening. We have expanded on the relevance of such further data collection in the text.  

Table one: Right hand section of the table is missing. Table formatting in general is poor, making it hard to read the table. Which species are in which enclosure.

Thank you for your comments here. We have simplified this table and ensured it is easier to read.

It would have been useful to observe the flamingo behaviour again after the zoo had been open for the same amount of time it had been closed. There are probably temporal effects to the behavioural responses. It doesn’t seem a “fair” comparison to observe them after they had become accustomed to no visitors, and then immediately after visitation resumed, without examining them again once they had become used to visitors. Observations immediately visitor restrictions commenced would have likewise been useful.

Figure 1 The legend is not particularly readable.

Thank you for the feedback. We have edited this figure for clarity.

Why the 3x 1.5h observation periods per day? How were these times selected?

This was dependent on what the zoos would allow given that we were working in a lockdown when it was difficult to travel and gain access. We were unable to access the zoo more frequently.

Paragraph line starting 194: do you mean visitor numbers into the zoo, or visitor numbers at the enclosure here?

Thank you, we have clarified this.

Line 205 Performance of behaviour?

Edited to time spent on behaviour

Paragraph starting line 205 This list of % is not very readable. Just summarise the major findings in the text and use the figure to show the detail.

Thank you for the suggestion. This paragraph has since been edited based on previous review comments.

Figure 3.1. Is the figure numbering correct?

Thank you for your comment, we tried to show each figure as part of the section it was in, i.e. Results is section 3, so 3.1, 3.2 but we have edited for simplicity to Figure 1, 2, 3 etc.

Figure 3.1 Quality of the text is poor – it is pixilated. Suggest using a graphics package to ensure high-quality output. Can you use various forms of shading/hatching rather than colour to distinguish the various bars?

Thank you for the comment. We do not have any pixilation in our  file, so we attach the actual image to the submission.

When are the animals fed relative to the observation periods?

Thank you for the comment. We have included details on their daily feeding schedule in the methods and in the text for discussion.

The results section is difficult to read; work on organising the results to make the major points clear, and on writing style to aid the reader. You need to summarise your findings and make the important information clear.

Thank you for the comment. We have organised the results by species and we have included the summary table at the end to provide the reader with clear explanation of the main findings. We have renamed each section of the results so the outcome variable is clear.

Table 3.1 Why the *** for significance codes when the p values are given?

Thank you for the comment. We have edited this.

Table 4 this summary table is unnecessary and is not the way to open the discussion. Use this table to help you organise the results section – your results section should make these questions and the corresponding answers clear.

Thank you for the comment. This table is the summary at the end of the results section, which  leads into the discussion. We have included a line to explain this in the discussion.

The first part to the discussion is really results.

Thank you for the feedback. We have edited this section to link to Table 4 at the end of the results.  

Line 392 This sentence doesn’t make sense.

Thank you for the suggestion. We have edited this sentence.

Chilean flamingos fed a lot less when there were visitors. Are there any data for body mass or body condition to examine the impact of this?

Thank you for the suggestion for the extra data. I am afraid we have no data on this as it was only an observational study and we were unable to gain access to extra records during a difficult time for the zoo to be operating. We have included a suggestion of looking at biometric measurements in the discussion as a useful research extension.

Reduction in feeding is not a negative impact?

Thank you for the comment. We do not believe so in this instance as the birds were responding more to the interaction between weather and visitors as therefore as they had the opportunity to forage ad lib throughout the day, any reduction in feeding during observation periods may well have been made up over night or in the evening. We have explained this point in the discussion.

The correlation of behaviour and temperature highlights the issue with limited sampling periods.

Thank you for the suggested discussion point. We have included this as a research extension in the discussion.

The length of the discussion could be reduced by 50%.

Thank you for the comment. We have had to expand on certain points in the discussion to accommodate the edits of all reviewers so it is tricky to shorten its length. However, we have tightened up the wording and reduced unnecessary description where needed. If you have specific suggestions for which parts to remove, we will gladly action these.

Line 480 Much of this section is really results. Beware of re-stating results in the discussion.

Thank you for your feedback. We have edited this section accordingly.

Formatting of references needs to be consistent (and in line with journal style). Latin names need to be in italics.

Thank you for the comment. We have formatted the reference as per the journal’s numbered style. Apologies for not doing this on the first submission.